# VEGF-C and Lymphatic Vessel Density in Tumor Tissue of Gastric Cancer: Correlations with Pathoclinical Features and Prognosis

**DOI:** 10.3390/cancers17091406

**Published:** 2025-04-23

**Authors:** Mariusz Szajewski, Maciej Ciesielski, Rafał Pęksa, Piotr Kurek, Michał Stańczak, Jakub Walczak, Jacek Zieliński, Wiesław Janusz Kruszewski

**Affiliations:** 1Department of Oncological Surgery, Gdynia Oncology Centre, 81-519 Gdynia, Poland; maciej.ciesielski@gumed.edu.pl (M.C.); piotr_kurek@gumed.edu.pl (P.K.); mstanczak@gumed.edu.pl (M.S.); jwalczak@szpitalepomorskie.eu (J.W.); wieslaw.kruszewski@gumed.edu.pl (W.J.K.); 2Department of Oncological Surgery, Faculty of Health Sciences with the Institute of Maritime and Tropical Medicine, Medical University of Gdansk, 80-210 Gdansk, Poland; 3Department of Pathomorphology, Faculty of Medicine, Medical University of Gdansk, 80-210 Gdansk, Poland; rafal.peksa@gumed.edu.pl; 4Department of Oncological Surgery, Faculty of Medicine, Medical University of Gdansk, 80-210 Gdansk, Poland; jaziel@gumed.edu.pl; 5Department of General and Oncological Surgery, Janusz Korczak Specialist Hospital, 76-200 Slupsk, Poland; 62nd Division of Radiology, Faculty of Health Sciences with the Institute of Maritime and Tropical Medicine, Medical University of Gdansk, 80-210 Gdansk, Poland

**Keywords:** VEGF-C, lymphangiogenesis, gastric cancer

## Abstract

The process of neoangiogenesis, which is associated with the progression of gastric cancer, has been widely recognized as a desirable target in the fight against this disease, leading to the incorporation of anti-angiogenic drugs into its treatment. The role of lymphangiogenesis in gastric cancer progression remains an area of intensive research, and the prognostic value of factors driving this process—which could potentially be promising therapeutic targets—is still unclear, especially in non-Asian populations. Based on a unique Polish cohort of radically operated gastric cancer patients without neoadjuvant chemotherapy, we confirm previous findings showing no correlation between VEGF-C expression or lymphatic vessel density and prognosis. However, in a novel discovery, we found that the Lauren intestinal type of gastric cancer is associated with VEGF-C overexpression, which may serve as an unfavorable prognostic factor in this subtype.

## 1. Introduction

Gastric cancer is the fifth most common malignant neoplasm and the fifth most common cause of death due to cancer in the world [1]. This has made it the subject of intensive research in order to improve the effects of its treatment. The progression of gastric cancer locally and beyond the primary site is possible due to the formation of a network of blood and lymphatic vessels (LVs) [2,3,4]. In the era of rapidly evolving immunotherapy, lymphangiogenesis—as a key regulatory mechanism in the tumor microenvironment—is attracting increasing attention [5,6]. LVs may promote the initiation of the immune response; however, they can also weaken the antitumor immune response, causing tumor immune evasion [5,7,8]. Tertiary lymphatic structures (TLS) within tumor tissue, as well as secondary lymphatic structures (SLS), are important biological markers of cancer progression and the response to oncological therapy. These structures are directly affected by VEGF-C, a member of the VEGF family that is considered a key factor in regulating tumor-associated lymphangiogenesis. LVs provide immune transport functions in TLS similar to those observed in the tumor microenvironment and lymph nodes [5]. The density of lymphatic vessels (LVD) within the tumor exceeds that observed in the surrounding healthy tissues [4,9,10]. VEGF-C is synthesized and secreted by cancer cells and tumor-associated macrophages that are stimulated by VEGF-C derived from tumor cells [5]. It has the ability to bind to and activate the VEGFR-3 receptors located on lymphatic endothelial cells (LECs), in addition to activating VEGFR-2, which is located on the vascular endothelium (VEC). The ultimate effect of VEGF-C, mediated through its described receptors, is the growth of LVs and blood vessels (VVs) accompanying the tumor [3,4,5,11,12,13,14,15]. The overexpression of VEGF-C seems to promote increased peritumoral lymphatic vessel density (LVD), which the authors associate with an increased risk of lymph node metastases. Increased LVD is believed to favor a worse prognosis of gastric cancer [9,11,16,17,18]. The results of many studies have indicated that the overexpression of VEGF and VEGF-C also contributes to a worse prognosis of gastric cancer [18,19,20,21]. However, one review study with a meta-analysis involving 3999 cases of gastric cancer in patients from Asia did not indicate any relationship between VEGF-C and prognosis [22]. The available literature does not clarify how the overexpression of VEGF-C in gastric cancer affects the pathoclinical parameters of the disease but suggests that it promotes unfavorable prognostic factors [10,23,24]. Similarly, it remains unclear how peritumoral LVD influences prognostic factors in gastric cancer and the prognosis itself, with contradictory results published [2,9,20]. This, along with the crucial role of lymphangiogenesis in gastric cancer biology and the paucity of publications in recent years on the prognostic value of VEGF-C expression, inspired us to investigate the prognostic value of VEGF-C overexpression and peritumoral LVD in gastric cancer using our own data.

## 2. Materials and Methods

This retrospective study included a group of 103 adult patients who underwent radical surgery for gastric adenocarcinoma at the Department of Surgical Oncology of the Medical University of Gdansk in the years 2006–2013. They received no neoadjuvant treatment before surgery. Gastric cancer was their first malignancy. Laboratory test preparations were made from sections of the primary tumor taken from formalin–paraffin blocks. This study was approved by the local Bioethics Committee (NKBBN/427/2014). The minimum follow-up period was 61 months. All deaths during follow-up were caused by cancer. The average survival time in the analyzed group was 50 months, with a median of 40 months. The pathoclinical parameters of the analyzed group of patients are presented in Table 1. The age of the patients was related to the day of surgery. The stage of cancer was determined based on the pTNM classification [25]. In the row referring to the location of the tumor, “cardia” indicates type III tumors according to the classification by Siewert et al. [26]. The “other location” category means tumors located in the body, on both curvatures, and in the pyloric part of the stomach. The immunohistochemical procedure and microscopic evaluation of the obtained preparations were performed at the Department of Pathomorphology at the Medical University of Gdansk.

Tissue material fixed in a 4% buffered formaldehyde solution and embedded in low-melting paraffin was used for immunohistochemical staining. Paraffin blocks were cut on a slide microtome into 4 µm-thick sections, which were placed on silanized glass slides. The sections were incubated in an incubator at 36 °C for 24 h. Immunohistochemical tests were performed in 103 cases of resectable gastric adenocarcinoma. The antibodies used had the following characteristics.

VEGF-C (c-20) sc 1881-Goat Polyclonal IgG (Santa Cruz Biotechnology Dallas, TX, USA): A primary antibody dilution of 1:100 was used, with an incubation time of 1 h. The Vector visualization system used was the Imm PRESS Reagent Kit with peroxidase anti-goat Ig. The procedure was performed manually.

Podoplanin D2-40 Mouse Monoclonal Antibody (ROCHE): This procedure was performed on a ROCHE Benchmark GX machine (ROCHE, Basel, Switzerland).

The cytoplasmic expression of VEGF-C in gastric adenocarcinoma cells was assessed using the semi-quantitative high score (HS) method. This method determined the intensity of the immunohistochemical reaction within a given case on a scale from 0 to 3 (0: no reaction; 1+: weak reaction; 2+: medium reaction; 3+: strong reaction), and the area of cells with a specific reaction, given as a percentage in relation to the entire surface of the examined lesion, as shown in Figure 1.

Based on the data obtained, the immunohistochemical reaction expression index was calculated according to the following formula:HS = 1 × a + 2 × b + 3 ×c
where a, b, and c represent the percentages of the lesion area occupied by cells with a specific reaction intensity, and × represents multiplication [27]. Tumors were defined as VEGF-C (+) if the HS index was >10 (over 10% of the lesion surface with a reaction of at least 1+). Tumors with an HS index of 10 were defined as VEGF-C (−).

Using monoclonal antibodies against podoplanin (D2-40), the peritumoral LVD was measured at the junction of the tumor and healthy tissue according to the methods of Weidner et al. [28]. First, under low microscope magnification (×100), areas with a high density of lymphatic vessels, so-called hot spots, were identified. Then, in the hot spots, under high microscope magnification (×200), the lymphatic vessels were counted, taking the average value of the measurements at three points as a measure of LVD. Each cluster of endothelial cells or individual vessel was regarded as a hot spot and counted as one microvessel. For the purpose of assessing the impact of LVD on prognosis, tumors were divided into two groups based on median vessel density. Figure 2 shows a microscopic image of labeled lymphatic vessels with the presence of tumor cell emboli.

Statistical analysis was performed using the tools of the STATISTICA package (data analysis software system), version 12 (www.statsoft.com, StatSoft, Inc. (2014)). The relationship between the expression of VEGF-C and LVD, as well as the pathoclinical and LVD parameters, was assessed using the Mann–Whitney U test. The relationship between the pathoclinical parameters of gastric adenocarcinoma and VEGF-C expression was assessed using Pearson’s chi-square test. One-way survival analysis was performed using the Kaplan–Meier method, and differences between the groups were verified with the log-rank test. The Cox’s proportional hazards model was used in multivariate analysis. The significance level for all calculations was 0.05.

## 3. Results

VEGF-C (+) tumors accounted for 73% (*n* = 75), and VEGF-C (−) tumors accounted for 27% (*n* = 28). The LVD in the study group, determined according to the method of Weidner et al., was 15 vessels/field of view (mean of 15, median of 15, and range of 0–48 vessels). The relationship between VEGF-C expression and the mean LVD is shown in Table 2.

An increased LVD was observed in cases with VEGF-C overexpression (*p* = 0.03). The relationships between VEGF-C expression in the VEGF-C (+) vs. VEGF-C (−) groups and between LVD and pathoclinical parameters of gastric adenocarcinoma are summarized in Table 3.

The intestinal type of gastric adenocarcinoma according to the Lauren classification is characterized by the overexpression of VEGF-C. A relationship between LVD and the location of the tumor in the stomach was observed. Postcardiac tumors showed significantly greater LVD (*p* = 0.04). The impact of LVD on the prognosis was assessed in the following groups: group A—tumors in which the LVD <15 vessels; group B—tumors in which the LVD ≥15 vessels. The results of the univariate survival analysis are presented in Table 4.

As expected, more advanced cases, with deeper infiltration of the gastric wall or with metastases in regional lymph nodes, clearly had a worse prognosis. However, the expression level of VEGF-C and the LVD did not affect the OS in the analyzed group (*p* = 0.8 and 0.1, respectively). A multivariate analysis of the entire dataset using a Cox model showed that the pT feature was an independent factor influencing survival time in the analyzed group (*n* = 103) (*p* = 0.02, CI 95% 0.1–1.67, HR 2.43). Survival analysis was also performed on a group of only Lauren intestinal-type cancers (*n* = 51). The univariate analysis showed that deeper infiltration of the gastric wall, the presence of metastases in regional lymph nodes, and generally more advanced staging were associated with worse prognosis, as was the overexpression of VEGF-C (Table 5). In the multivariate analysis of this group of patients, only VEGF-C overexpression showed an independent impact on worse prognosis, with a borderline significance value (*p* = 0.05, CI 95%: −0.03–3.99, HR: 7.2).

## 4. Discussion

Cytoplasmic expression of VEGF-C in gastric cancer cells has been observed in 6–75% of gastric cancer cases [9,10,11,16,19,21,22,24,29]. The immunohistochemical method is most commonly used to address this expression [2,9,10,13,19,22]. Various markers specific for lymphatic endothelial cells are used for the immunohistochemical detection of LVs [2,9,10,13,19,22]. Among these, the D2-40 monoclonal antibody against podoplanin is particularly effective in detecting lymphatic vessels; therefore, we used it in our study [2,9,10,24]. A distinction is made between intratumoral and peritumoral LVs. However, in invasive gastric cancer, the latter are particularly responsible for lymphogenous cancer progression. For this reason, we assessed the density of peritumoral LVs [4,9,12,16]. Similar to other authors, we used the method of Weidner et al. to assess LVD [9,28,30]. According to the methods described above, the average LVD value in VEGF-C (+) gastric cancers ranged from 10 to 19 vessels [9,17,31,32,33]. We obtained similar results (mean of 16; median LVD of 15 vessels) in our study. In the studies of many authors from Asia, the overexpression of VEGF-C in gastric cancer is strongly associated with the occurrence of unfavorable prognostic factors, such as metastases in regional lymph nodes or more advanced stages of the disease [11,16,17,18,19,21,23,24]. Additionally, *VEGF-C* has been included in the group of seven key genes with prognostic value in gastric cancer. *VEGF-C* upregulation is expected to lead to overexpression of the VEGF-C protein, which is associated with worse prognosis [34]. Furthermore, Italian authors showed that higher levels of VEGF-C protein in serum before surgery indicated worse prognosis of gastric cancer patients [35]. This was also clearly indicated in a review study with meta-analysis by Liu et al. based on the data of 4974 gastric cancers [19]. However, in a meta-analysis by Peng et al. [22], published in parallel with the meta-analysis by Liu et al. [19], there was no relationship between the level of VEGF-C in the primary tumor and prognosis. Liu et al. [19] assessed tumor VEGF-C based on studies only from Asia. In their assessment of the relationship between VEGF-C and OS (*n* = 1398), Peng et al. [22] also used reports by Asian authors only [22]. In our study based only on non-Asian patients, we also did not find a correlation between VEGF-C overexpression in the primary tumor and the prognosis of gastric cancer. Other Polish authors drew similar conclusions in their research [20]. However, in the case of intestinal-type cancer, VEGF-C overexpression was associated with worse prognosis, according to our data. Many authors have shown positive correlations between VEGF-C overexpression and lymph node metastases, higher clinical stage of the disease, or shorter OS. However, when examining the relationship between VEGF-C or LVD and the Lauren type, they did not find a significant relationship between VEGF-C expression and the Lauren type [11,24], nor with the degree of cancer differentiation [36], regardless of the VEGF-C protein detection method.

We have shown that increased peritumoral LVD is clearly associated with VEGF-C overexpression (*p* = 0.03, Table 2), which is consistent with the results of numerous studies [10,16,17,32,33]. Authors from Asia have demonstrated that increased LVD is associated with worse prognosis. According to the data of 1072 cases of gastric cancer, the independent negative prognostic value of this feature was indicated [9]. This was also confirmed in the work by Chinese authors who showed a positive correlation between a higher amount of VEGF-C mRNA and increased LVD in more advanced cases [36]. However, several studies did not demonstrate a relationship between VEGF-C and the LVD [11,20,37], one of which was conducted in Poland [20].

We did not find an association between VEGF-C in pericardia tumors and the LVD. However, tumors with a lower LVD were significantly more frequent in this location than those located more peripherally. In contrast, based on pericardium cancer data, authors from Asia showed a significant relationship between VEGF-C overexpression, increased LVD, and worse cancer prognosis, highlighting higher peritumoral VEGF-C and LVD overexpression [17].

According to our data, the Lauren intestinal type of cancer was significantly more often accompanied by VEGF-C overexpression, and cases with VEGF-C overexpression showed worse prognosis in univariate analysis. At *p* = 0.05, VEGF-C overexpression in the multivariate analysis indicated an independent prognostic value of VEGF-C in intestinal-type cancer. There are significant differences in the biology of gastric cancer depending on the Lauren type, determined at the molecular level, with the diffuse type of cancer exhibiting a more invasive phenotype, which predisposes to intraperitoneal dissemination [3,11,18,38]. It was shown that in diffuse-type tumors, invasive behavior was characterized by the upregulation of epithelial–mesenchymal transition-related genes, and in intestinal-type tumors, proliferative behavior was characterized by the upregulation of cell cycle-related genes. Therefore, molecular pathways characterizing intestinal-type tumors, together with other cancer-related pathways, are involved in cell cycle regulation and the response to growth factors and hormones (EGFR, estradiol, and progesterone) [11,39]. This may be the reason for the significantly more frequent overexpression of VEGF-C in intestinal cancer in our data (Table 3). Similar to other authors, we also noticed a smaller percentage of women in the group with intestinal-type cancer (22%) than in the group with diffuse-type cancer (37.5%) (Table 1). It has been suggested that this may be the result of the protective effect of high endogenous estrogen exposure [39]. Taking into account the histological Lauren type, the risk of the type of dissemination can be predicted more precisely by additionally determining the genotype of the cancer [38]. We found no reports on a relationship between VEGF-C and the promotion of a specific route of cancer spread depending on the histological type. Although it has been shown that intestinal cancer spreads mainly through blood vessels due to the overexpression of VEGF-A, accompanied by an increased LVD in the primary tumor [3], we did not demonstrate the effect of LVD on the prognosis of intestinal-type cancer.

When assessing the entire dataset, we did not find any relationship between the expression of VEGF-C or LVD and age, sex, pT, pN, or pTNM. VEGF-C also did not correlate with the location of the cancer, while pericardia tumors were significantly more often characterized by a lower density of LVs (Table 5). The results of a study from China clearly indicated worse prognosis for gastric cancer with the simultaneous overexpression of VEGF-C and VEGF-A, regardless of the Lauren type. The combined occurrence of these two factors is expected to promote increased VVD and LVD in the tumor, correlating with a worse prognosis. The lack of correlation with the prognosis and prognostic factors in cases of VEGF-A (−)/VEGF-C (+) and VEGF-A (+)/VEGF-C (−) suggests that their type of expression is mutually dependent and jointly affects the progression of gastric cancer. Therefore, these markers should be assessed together for predictive purposes. The authors demonstrated in a mouse model that silencing the mutual overexpression of these markers promoted the regression of gastric cancer [13]. This may explain the heterogeneity of the results obtained by different authors examining the prognostic value of VEGF-C alone in gastric cancer. It also indicates the value of searching for VEGF-C together with VEGF-A and treating them both as promising prognostic factors and potentially as predictive factors for therapy. Similarly, this can explain the significant heterogeneity of the results of studies assessing the prognostic value of VEGF-C and peritumoral LVD in gastric cancer. VEGF-C, rather than the LVD, seems to be more important as a potential prognostic factor in gastric cancer, as it affects a broader spectrum of cells in the tumor microenvironment beyond just lymphatic endothelial cells. While VEGF-C is essential for lymphatic system development and plays a crucial role in tumor lymphangiogenesis, it also co-creates the axis VEGF-C/VEGFR which affects multiple important cell processes in cancer progression, such as proliferation, invasion, and resistance to chemotherapy. Inhibiting VEGF-C signaling is recognized as a potentially valuable treatment method in oncology [5,6,12].

The foremost limitation of the present study is the relatively small patient cohort derived from a single institution, potentially introducing selection bias, compounded by its retrospective design. Nonetheless, this methodology was indispensable to compile a consecutive series of patients representing various stages of gastric cancer who had not undergone neoadjuvant chemotherapy, a treatment now deemed mandatory in more advanced cases.

## 5. Conclusions

Peritumoral overexpression of VEGF-C in primary gastric cancer tumors is associated with increased LVD. The Lauren intestinal type of cancer is associated with VEGF-C overexpression, and the overexpression of VEGF-C in intestinal-type gastric is associated with worse prognosis.

## Figures and Tables

**Figure 1 cancers-17-01406-f001:**
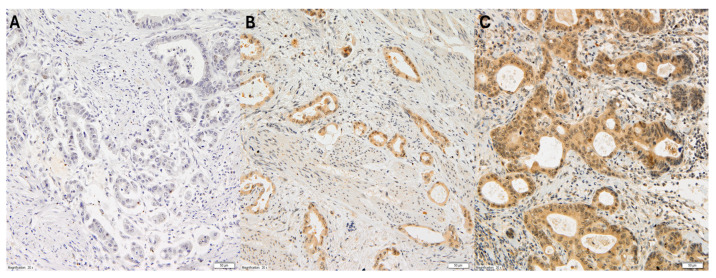
Representative immunohistochemical staining of VEGF-C in gastric cancer at 20× magnification. The three panels (**A**–**C**) illustrate varying levels of VEGF-C expression: (**A**) no detectable staining (score 0), representing VEGF-C-negative tumor cells; (**B**) weak cytoplasmic staining (score +1), observed in a subset of tumor cells; (**C**) moderate to strong cytoplasmic staining (score +2), present in a significant proportion of tumor cells.

**Figure 2 cancers-17-01406-f002:**
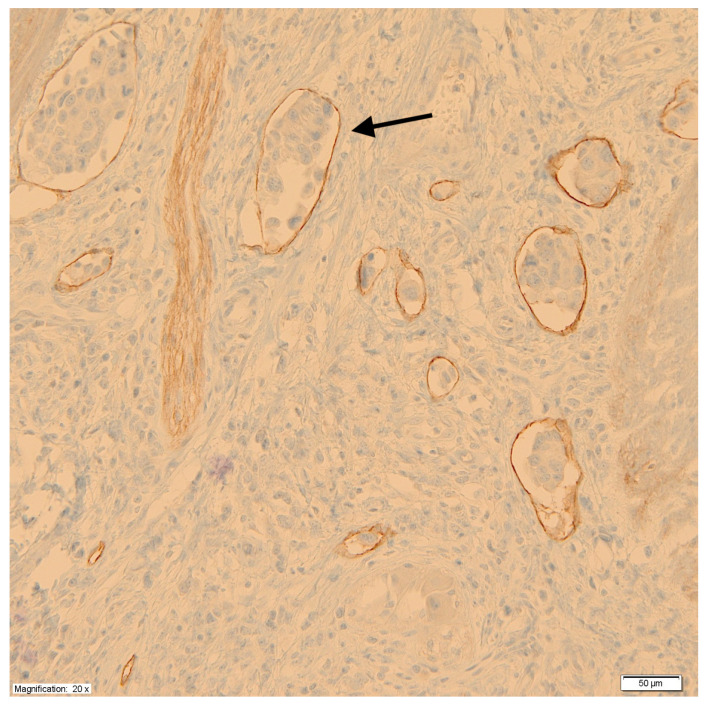
Labeled lymphatic vessels with the presence of tumor cell emboli (black arrow).

**Table 1 cancers-17-01406-t001:** Clinicopathological characteristics of 103 patients with gastric adenocarcinoma.

Parameter		No. of Patients	%
Age (median: 62 years)	<62 ≥62	48 55	47 53
Gender	Female Male	31 72	30 70
Depth of invasion (pT)	T1 (a + b) T2 T3 T4 (a + b)	9 (1 + 8) 17 53 24 (19 + 5)	8.5 16.5 51.5 23.5
Regional lymph nodes (pN)	N0 N + (N1+ N2 + N3 _a + b_)	26 77 (23 + 22 + 32 _22 + 10_)	25 75
Stage (pTNM)	I (A + B) II (A + B) III (A + B + C)	15 (6 + 9) 31 (12 + 19) 57 (24 + 22 + 11)	14.5 30 55.5
Histological type (Lauren)	Intestinal - Females: 11 (22%) - Males: 40 (78%) Diffuse - Females: 12 (37.5%) - Males: 20 (62.5%) Mixed - Females: 8 (40%) - Males: 12 (60%)	51 32 20	49.5 31 19.5
Location of tumor	Cardia Other location	32 71	31 69

**Table 2 cancers-17-01406-t002:** Relationship between LVD and expression of VEGF-C (Mann–Whitney U test).

Parameter	VEGF-C (−) Mean (SD)	VEGF-C (+) Mean (SD)	*p* Value
LVD	12 (±8)	16 (±9)	0.03

**Table 3 cancers-17-01406-t003:** Relationships among expression of VEGF-C (Pearson’s chi-square test), LVD (Mann–Whitney U test *) and clinicopathological parameters.

Parameter		VEGF-C (−) %	VEGF-C (+) %	*p* Value	LVD Median	*p* Value *
Age (median: 62 years)	<62 ≥62	35 20	65 80	0.07	15 12	0.4
Gender	Female Male	29 26	71 74	0.7	15 14	0.9
Depth of invasion (pT)	T1 + T2 T3 + T4	27 27	73 73	0.9	15 13	0.08
Regional lymph nodes (pN)	N (−) N (+)	27 27	73 73	0.9	15 14	0.2
Stage (pTNM)	I + II III	26 28	74 72	0.8	15 13	0.1
Histological type (Lauren)	Intestinal Diffuse + Mixed	12 42	88 58	<0.001	13 15	0.4
Location of tumor	Cardia Other location	19 31	81 69	0.1	10 15	0.04

**Table 4 cancers-17-01406-t004:** The influence of selected clinicopathological parameters on overall survival (*n* = 103).

Parameter		Overall Survival Probability	Log-Rank *p* Value
Expression of VEGF-C	VEGF-C (−) VEGF-C (+)	0.32 0.32	0.8
LVD	A B	0.25 0.38	0.1
Age (median: 62 years)	<62 ≥62	0.37 0.27	0.06
Gender	Female Male	0.32 0.32	0.9
Depth of invasion (pT)	T1 + T2 T3 + T4	0.61 0.22	<0.001
Regional lymph nodes (pN)	N (−) N (+)	0.58 0.23	<0.001
Stage (pTNM)	I + II III	0.48 0.19	<0.001
Histological type (Lauren)	Intestinal Diffuse + Mixed	0.41 0.23	0.06
Location of tumor	Cardia Other location	0.25 0.35	0.4

**Table 5 cancers-17-01406-t005:** The influence of selected clinicopathological parameters on overall survival (Lauren intestinal type, *n* = 51).

Parameter		*n*	Overall Survival Probability	Log-Rank *p* Value
Expression of VEGF-C	VEGF-C (−) VEGF-C (+)	6 45	0.83 0.36	0.01
LVD	A B	39 12	0.31 0.52	0.1
Age (median: 62 years)	<62 ≥62	24 27	0.46 0.37	0.2
Gender	Female Male	11 40	0.45 0.40	0.7
Depth of invasion (pT)	T1 + T2 T3 + T4	12 39	0.67 0.33	0.01
Regional lymph nodes (pN)	N (−) N (+)	14 37	0.64 0.32	0.02
Stage (pTNM)	I + II III	24 27	0.54 0.30	0.01
Location of tumor	Cardia Other location	21 30	0.33 0.47	0.5

## Data Availability

The raw data supporting the conclusions of this article will be made available by the authors upon request.

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
