# Peer review of "VEGF-C and Lymphatic Vessel Density in Tumor Tissue of Gastric Cancer: Correlations with Pathoclinical Features and Prognosis"

_cancers, 2025, doi:10.3390/cancers17091406_

Round 1

Reviewer 1 Report

Comments and Suggestions for Authors

The author raised several questions , such as how overexpression of VEGF-C in gastric cancer cells affects the pathoclinical parameters of this disease. And then assessed the relationship of VEGF-C and LVD with pathoclinical factors of potential prognostic value and with the survival of gastric cancer patients. However, the results in this study do not adequately address these questions. There are still a number of key issues that need to be tackled.

  1. It would be better to present he VEGFC immunohistochemical staining within the results section, showing images that reflect various scoring criteria rather than a single positive staining image. Additionally, the quality of the images requires improvement.
  2. In the methods section, how was the lymphatic vessel counted? It should be clearly described.
  3. The selective influence of VEGFC on Lauren's intestinal-type gastric cancer needs a more thorough analysis and explanation to elucidate the underlying reasons for this specificity.
  4. The discussion should address why VEGFC, rather than lymphatic vessel density themselves, exhibits a correlation with cancer progression. It should be considered that VEGFC exerts its influence beyond lymphatic endothelial cells, potentially affecting a broader range of cells in the tumor microenvironment.
Comments on the Quality of English Language

The introduction and discussion sections should be improved to ensure that the logic is clear and the language is concise.

Author Response

Dear Reviewer,

Thank you for your time and attention and for allowing us to improve the text of our work. We have tried to revise the text following your remarks and recommendations.

Comments and Suggestions for Authors

The author raised several questions, such as how overexpression of VEGF-C in gastric cancer cells affects the pathoclinical parameters of this disease. And then assessed the relationship of VEGF-C and LVD with pathoclinical factors of potential prognostic value and with the survival of gastric cancer patients. However, the results in this study do not adequately address these questions. There are still a number of key issues that need to be tackled.

  1. Comments 1: It would be better to present he VEGFC immunohistochemical staining within the results section, showing images that reflect various scoring criteria rather than a single positive staining image. Additionally, the quality of the images requires improvement.

Response1:The presentation of VEGFC immunohistochemical staining is now in the Results section, and images display various scoring criteria. We have changed the figures.

  1. Comments 2: In the methods section, how was the lymphatic vessel counted? It should be clearly described.

Response 2: The description on how was the lymphatic vessel counted has been added – lines 153-155

  1. Comments 3:The selective influence of VEGFC on Lauren's intestinal-type gastric cancer needs a more thorough analysis and explanation to elucidate the underlying reasons for this specificity.

Response 3: We have inserted new text in the Discussion on this, lines 272-278

  1. Comments 4: The discussion should address why VEGFC, rather than lymphatic vessel density themselves, exhibits a correlation with cancer progression. It should be considered that VEGFC exerts its influence beyond lymphatic endothelial cells, potentially affecting a broader range of cells in the tumor microenvironment.

Response 4: We have inserted new text in the Discussion on this, lines 308-315

Comments on the Quality of English Language

The introduction and discussion sections should be improved to ensure that the logic is clear and the language is concise.

We have made changes to the chapters.

We also sent our manuscript to the MDPI editorial office to improve the English language.

Reviewer 2 Report

Comments and Suggestions for Authors

This manuscript has investigated VEGF-C and podoplanin in a small cohort of gastric cancers. The study is not novel in its design and several equivalent studies have been reported previously, with similar findings. The novelty of the study is related to the study population, which is not well-represented in the literature. The study has been performed and analysed appropriately, with scoring of immunostained slides consistent with accepted pathology practice and clear presentation of results tables. A major problem with the manuscript is the over-interpretation of results and the length of the Discussion in relation to the study size and amount of results.

  1. English language editing will be required in order to correct a number of grammatical and syntax (word usage) errors throughout the text.
  2. The microscopy images in Figures 1 and 2 are very faint and lacking contrast. The magnification could be increased to improve their clarity.
  3. Association is not the same as causation. In the Abstract and elsewhere in the manuscript, the authors state that VEGF-C overexpression “promoted” increased LVD, however there is no evidence either from the authors or from the published literature to indicate a causative relationship. Based on the study results, the most that the authors could state is that VEGF-C overexpression “was associated with” increased LVD. This type of overstatement of results should be corrected wherever it occurs in the text.
  4. Could the authors please add the group numbers to Table 5 (e.g. numbers of VEGF-C(-)/VEGF(+) tumours, number of male/female patients, etc)? Due to the small size of this group overall (51 patients), the ability to derive definitive conclusions is limited and should be stated as such.
  5. In line 206 of the manuscript, the authors state that the mean LVD in their study was 16. However, earlier in the manuscript in the Results section (lines 157-158) a figure of 15 is stated. Which is correct?
  6. Because the design of the study is not novel, the methods are limited in number and scope, and the cohort is small, the Discussion section is disproportionately long and should be shortened to better represent the content of the manuscript.
Comments on the Quality of English Language
  1. English language editing will be required in order to correct a number of grammatical and syntax (word usage) errors throughout the text.

Author Response

Dear Reviewer,

Thank you for your time and attention and for allowing us to improve the text of our work. We have tried to revise the text following your remarks and recommendations as below.

Comments and Suggestions for Authors

This manuscript has investigated VEGF-C and podoplanin in a small cohort of gastric cancers. The study is not novel in its design and several equivalent studies have been reported previously, with similar findings. The novelty of the study is related to the study population, which is not well-represented in the literature. The study has been performed and analysed appropriately, with scoring of immunostained slides consistent with accepted pathology practice and clear presentation of results tables. A major problem with the manuscript is the over-interpretation of results and the length of the Discussion in relation to the study size and amount of results.

  1. Comments 1: English language editing will be required in order to correct a number of grammatical and syntax (word usage) errors throughout the text.

Response 1: The text correction has been made. We also sent our manuscript to the MDPI editorial office to improve the English language.

  1. Comments 2: The microscopy images in Figures 1 and 2 are very faint and lacking contrast. The magnification could be increased to improve their clarity.

Response 2: Please, see new Figures included

  1. Comments 3: Association is not the same as causation. In the Abstract and elsewhere in the manuscript, the authors state that VEGF-C overexpression “promoted” increased LVD, however there is no evidence either from the authors or from the published literature to indicate a causative relationship. Based on the study results, the most that the authors could state is that VEGF-C overexpression “was associated with” increased LVD. This type of overstatement of results should be corrected wherever it occurs in the text.

Response 3: The recommended correction has been made

  1. Comments 4: Could the authors please add the group numbers to Table 5 (e.g. numbers of VEGF-C(-)/VEGF(+) tumours, number of male/female patients, etc)? Due to the small size of this group overall (51 patients), the ability to derive definitive conclusions is limited and should be stated as such.

Response 4: Please, see additional data in Table 1 and Table 5

  1. Comments 5: In line 206 of the manuscript, the authors state that the mean LVD in their study was 16. However, earlier in the manuscript in the Results section (lines 157-158) a figure of 15 is stated. Which is correct?

Response 5: The mean and median for LVD in the whole material is 15 vessels. Mean of 16 vessels relates to cases with VEGF-C (+) (Table 2).

Correction in the text has been done, lines 225-226

Comments 6: Because the design of the study is not novel, the methods are limited in number and scope, and the cohort is small, the Discussion section is disproportionately long and should be shortened to better represent the content of the manuscript.

Response 6: Discussion section has been shortened

Comments on the Quality of English Language

  1. English language editing will be required in order to correct a number of grammatical and syntax (word usage) errors throughout the text.

The text correction has been made. We also sent our manuscript to the MDPI editorial office to improve the English language

Reviewer 3 Report

Comments and Suggestions for Authors

The introduction  parts are quite long. VEGF cannot be consider a specific marker of any case of gastric tumor invasivity, because the lymphatic apparatus of each case is not its unique characteristic, mainly in the current era of improved knowledge of lymphocytes as immunological agent. In addition, you do not discus the role of secondary or tertiary lymphatic organ.  In the discussion these point are to be mentioned. 

Author Response

Dear Reviewer,

Thank you for your time and attention and for allowing us to improve the text of our work. We have tried to revise the text following your remarks and recommendations as below.

Comments and Suggestions for Authors

Comments 1: The introduction  parts are quite long.

Response 1: The Introduction chapter has been shortened

Comments 2: VEGF cannot be consider a specific marker of any case of gastric tumor invasivity, because the lymphatic apparatus of each case is not its unique characteristic, mainly in the current era of improved knowledge of lymphocytes as immunological agent. In addition, you do not discus the role of secondary or tertiary lymphatic organ.  In the discussion these point are to be mentioned. 

Response 2: Please, see some changes throughout the text we put in relation to Your remarks as above.

Lines in the text: 59 – 72, 229 – 234, 304-315

Reviewer 4 Report

Comments and Suggestions for Authors

The article entitled “VEGF-C and lymphatic vessel density in tumor tissue of gastric cancer: correlations with pathoclinical features and prognosis by Mariusz Szajewski, et al. demonstrated that peritumoral overexpression of VEGF-C in the primary gastric cancer tumor promotes increased LVD and overexpression of VEGF-C in Lauren’s intestinal type gastric cancer predisposes to a worse prognosis.

This study has some value, however, the mention about the reason why the assessment of the relationship of VEGF-C and LVD is superior to other prognostic factors is slightly faint. As a result, the authors present a limited interest.

Thus, there are areas that need to be improved.

Major comments

In “Discussion”, authors didn’t sufficiently describe superior points regarding assessment of VEGF-C expression and LVD. Please explain its advantage over other conventional prognostic predictors. Furthermore, authors should mention future prospects about deep assessment of VEGF-C expression and LVD.

Minor comments

  • Describes in “Discussion”; Line 273-293 should summarize in one paragraph. Additionally, describes in “Discussion”; Line 223-227 and Line 294-299 is important part in this report. Authors should summarize those in one paragraph. There is not enough mention about clinical significance of assessment of VEGF-C expression and LVD as potential biomarker.
  • Describes in “Introduction”; Line 53-89 and in “Discussion”; Line 190-223 and Line 228-247, are somewhat redundant and needs shortening.
Comments on the Quality of English Language

The English could be improved to more clearly express the research.

Author Response

Dear Reviewer,

Thank you for your time and attention and for allowing us to improve the text of our work. We have tried to revise the text following your remarks and recommendations as below.

Comments and Suggestions for Authors

Comments 1: The article entitled “VEGF-C and lymphatic vessel density in tumor tissue of gastric cancer: correlations with pathoclinical features and prognosis” by Mariusz Szajewski, et al. demonstrated that peritumoral overexpression of VEGF-C in the primary gastric cancer tumor promotes increased LVD and overexpression of VEGF-C in Lauren’s intestinal type gastric cancer predisposes to a worse prognosis.

This study has some value, however, the mention about the reason why the assessment of the relationship of VEGF-C and LVD is superior to other prognostic factors is slightly faint. As a result, the authors present a limited interest.

Thus, there are areas that need to be improved.

Response 1: We have tried to emphasize in the text the purposefulness of studying VEGF-C and LVD as a key factors of lymphangiogenesis so vital for tumor microenvironment, as this process is drawing more and more attention of scientists in contemporary era of immunotherapy in oncology.

Comments 2: In “Discussion”, authors didn’t sufficiently describe superior points regarding assessment of VEGF-C expression and LVD. Please explain its advantage over other conventional prognostic predictors. Furthermore, authors should mention future prospects about deep assessment of VEGF-C expression and LVD.

Response 2: Please, see lines 59 – 71, 87, 229 – 234, 304-315

Comments 3: Describes in “Discussion”; Line 273-293 should summarize in one paragraph. Additionally, describes in “Discussion”; Line 223-227 and Line 294-299 is important part in this report. Authors should summarize those in one paragraph. There is not enough mention about clinical significance of assessment of VEGF-C expression and LVD as potential biomarker.

Response 3: Describes have been summarized in accordance with the recommendations, lines: 250-258

Comments 4: Describes in “Introduction”; Line 53-89 and in “Discussion”; Line 190-223 and Line 228-247, are somewhat redundant and needs shortening

Response 4: These parts of text have been shortened

Round 2

Reviewer 1 Report

Comments and Suggestions for Authors

All my questions have been answered.

Author Response

Dear Reviewer,

On behalf of all the authors, I would like to sincerely thank you for your valuable time, thoughtful comments, and constructive suggestions regarding our manuscript entitled " VEGF-C and lymphatic vessel density in tumor tissue of gastric cancer: correlations with pathoclinical features and prognosis" (Manuscript ID: 3519225).

We greatly appreciate your careful review and insightful feedback, which have helped us to improve the quality and clarity of our work. We have carefully addressed all your comments and revised the manuscript accordingly

Thank you once again for your effort and support in the evaluation of our study.

Sincerely,
Piotr Kurek, on behalf of all authors
Department of Oncological Surgery, Gdynia Oncology Centre, Gdynia, Poland

piotr_kurek@gumed.edu.pl

Reviewer 2 Report

Comments and Suggestions for Authors

The authors have made many amendments according to reviewers’ comments. The following are some minor additional comments.

  1. Abstract, line 50: “The Lauren intestinal type of cancer predisposes individuals to VEGF-C overexpression” should be “The Lauren intestinal type of cancer is associated with VEGF-C overexpression”. (VEGF-C isn’t overexpressed in ‘individuals’, it’s overexpressed in the tumours, and the term ‘predisposes’ is not applicable to this type of study).
  2. The image in Figure 1D is not very convincing (strong VEGF-C immunostaining in 5% of tumour cells). It is suggested that this panel is removed (Figure 1A, B and C are good).
  3. On page 4 lines 127-128, the authors use the terminology “the area of cells with a specific reaction”. Does this mean area of tumour cells?
  4. The authors describe scoring of VEGF-C immunostaining according to percentages of tumour (?) cells with weak, moderate or strong immunoreactivity, however their designation of VEGF-C positive cells is simply based on more than 10% positively stained tumour cells, a method consistent with previous studies. If this is the case, then reference to the alternative method of scoring can be deleted.
  5. In the Discussion section, (line 232), the authors mention “blood serum”. This should be “serum”.
  6. The phrase “appears to predispose to” is not appropriate in this study and should be replaced with “was associated with”. Apart from the Abstract, there are 3 other instances that I have found in the Discussion page line 243, and Conclusions lines 318 & 320.
  7. The closing statement of the Conclusions is an overstatement of the findings of this manuscript. There were only 6 cases of VEGF-C(-) tumours of Lauren intestinal type and the very small number is insufficient to make this type of conclusion. (It is not suitable as a conclusion because the result is not backed by sufficient data).
  8. The authors need to add a section on limitations of their study to the Discussion. Although results of the study may be useful in meta-analyses and to repeat findings of other authors, the small number of patients limits its reliability and utility.

Author Response

Dear Reviewer,

On behalf of all the authors, I would like to sincerely thank you for your valuable time, thoughtful comments, and constructive suggestions regarding our manuscript entitled " VEGF-C and lymphatic vessel density in tumor tissue of gastric cancer: correlations with pathoclinical features and prognosis" (Manuscript ID: 3519225).

We greatly appreciate your careful review and insightful feedback, which have helped us to improve the quality and clarity of our work. We have carefully addressed all your comments and revised the manuscript accordingly. Here are responses to each of your points ;

Comments 1: Abstract, line 50: “The Lauren intestinal type of cancer predisposes individuals to VEGF-C overexpression” should be “The Lauren intestinal type of cancer is associated with VEGF-C overexpression”. (VEGF-C isn’t overexpressed in ‘individuals’, it’s overexpressed in the tumours, and the term ‘predisposes’ is not applicable to this type of study).

Response 1:  This part of text has been changed, line 49

Comments 2: The image in Figure 1D is not very convincing (strong VEGF-C immunostaining in 5% of tumour cells). It is suggested that this panel is removed (Figure 1A, B and C are good).

Response 2: The image has been modified according to the suggestion.

Comments 3: On page 4 lines 127-128, the authors use the terminology “the area of cells with a specific reaction”. Does this mean area of tumour cells?

Response 3: Yes, by “the area of cells with a specific reaction,” we specifically refer to tumour cells. Only neoplastic epithelial cells were evaluated for immunohistochemical staining intensity and the percentage of positive cells.

Comments 4: The authors describe scoring of VEGF-C immunostaining according to percentages of tumour (?) cells with weak, moderate or strong immunoreactivity, however their designation of VEGF-C positive cells is simply based on more than 10% positively stained tumour cells, a method consistent with previous studies. If this is the case, then reference to the alternative method of scoring can be deleted.

Response 4: We thank the Reviewer for this valuable comment. Initially, VEGF-C immunostaining was assessed in detail using the H-score method, which incorporates both the intensity of staining and the percentage of positive tumour cells. This approach allowed us to capture the full spectrum of expression without predefining a cut-off value. However, for the purpose of statistical analysis and to ensure comparability with previous studies, we subsequently defined VEGF-C positivity as immunoreactivity present in more than 10% of tumour cells, in accordance with the published literature.

Comments 5: In the Discussion section, (line 232), the authors mention “blood serum”. This should be “serum”.

Response 5: It has been changed, line 227

Comments 6: The phrase “appears to predispose to” is not appropriate in this study and should be replaced with “was associated with”. Apart from the Abstract, there are 3 other instances that I have found in the Discussion page line 243, and Conclusions lines 318 & 320.

Response 6: It has been changed - lines: 237, 319-321

Comments 7: The closing statement of the Conclusions is an overstatement of the findings of this manuscript. There were only 6 cases of VEGF-C(-) tumours of Lauren intestinal type and the very small number is insufficient to make this type of conclusion. (It is not suitable as a conclusion because the result is not backed by sufficient data).

Comments 8: The authors need to add a section on limitations of their study to the Discussion. Although results of the study may be useful in meta-analyses and to repeat findings of other authors, the small number of patients limits its reliability and utility.

Response 7 and 8: The text about study limitations has been added – lines: 310 – 315.

Thank you once again for your effort and support in the evaluation of our study.
We hope that the revised version meets your expectations.

Sincerely,
Piotr Kurek, on behalf of all authors
Department of Oncological Surgery, Gdynia Oncology Centre, Gdynia, Poland

piotr_kurek@gumed.edu.pl

Reviewer 3 Report

Comments and Suggestions for Authors

The text has been improved and now clearer. 

Author Response

(The authors gave the same response as above.)
